# MULTI-HEAD RAG: SOLVING MULTI-ASPECT PROBLEMS WITH LLMS

## ABSTRACT

Retrieval Augmented Generation (RAG) enhances the abilities of Large Language Models (LLMs) by enabling the retrieval of documents into the LLM context to provide more accurate and relevant responses. Existing RAG solutions do not focus on queries that may require fetching multiple documents with substantially different contents. Such queries occur frequently, but are challenging because the embeddings of these documents may be distant in the embedding space, making it hard to retrieve them all. This paper introduces Multi-Head RAG (MRAG), a novel scheme designed to address this gap with a simple yet powerful idea: leveraging activations of Transformer's multi-head attention layer, instead of the decoder layer, as keys for fetching multi-aspect documents. The driving observation is that different attention heads learn to capture different data aspects. Harnessing the corresponding activations results in embeddings that represent various facets of data items and queries, improving the retrieval accuracy for complex queries. We provide an evaluation methodology and metrics, multi-aspect datasets, and real-world use cases to demonstrate MRAG's effectiveness. We show MRAG's design advantages over 18 RAG baselines, empirical improvements of up to 20% in retrieval success ratios, and benefits for downstream LLM generation. MRAG can be seamlessly integrated with existing RAG frameworks and benchmarks.

## 1 INTRODUCTION

Retrieval-Augmented Generation (RAG) (Lewis et al., 2020) emerged as a promising remedy for several key limitations of Large Language Models (LLMs). By decoupling knowledge from model weights, RAG reduces the risk of leaking confidential data (Yan et al., 2025), a critical concern when training on sensitive corpora. It also mitigates hallucinations (Huang et al., 2025b) by grounding LLM outputs in retrieved, verifiable information. The core mechanism involves augmenting a generative LLM with a retrieval module that fetches relevant passages from an external corpus in response to a query. Rather than relying solely on static, parametric knowledge, RAG dynamically incorporates retrieved content into the model's context, enabling more accurate and up-to-date responses. While early RAG systems required training task-specific retrievers and readers (Humeau et al., 2020), the current trend favors lightweight, in-context learning (ICL) approaches (Gao et al., 2024), which avoid the cost and complexity of retraining and allow for rapid knowledge updates without modifying the underlying LLM parameters.

A RAG pipeline consists of two main stages: data preparation and query execution. In the preparation stage, a vector database (DB) is constructed by embedding a collection of documents and storing these embeddings alongside their associated content. At inference time, the query is similarly embedded, and nearest-neighbor search retrieves the most relevant data items, which are then passed to the LLM for final answer generation. Ongoing developments in RAG have led to variety of different RAG designs (Gao et al., 2024).

Yet, no existing RAG method or benchmark explicitly targets *multi-aspectual* problems, that is, queries requiring the integration of multiple, semantically distinct aspects. For example, answering "What car did Alexander the Great drive?" (assuming no historical pretraining) requires retrieving unrelated documents on Alexander the Great and on car manufacturing, whose embeddings may lie far apart in the vector space. Such multi-aspect queries are common in industrial settings, as confirmed by extensive discussions with our industry collaborators and further supported by our analysis of over 35 industry reports (details are in Appendix A; we considered accident prevention,

healthcare, airport management, and others). For example, in a chemical plant accident, determining the cause might require accessing diverse and confidential documents related to worker psychology (*"Was it mismanagement?"*), equipment records (*"Was a part outdated or rusty?"*), weather conditions (*"Were there power spikes due to a storm?"*), or even microclimate (*"Was prolonged humidity a factor?"*). As shown in Section 5, such cases have been unaddressed by modern RAG schemes and benchmarks (Chen et al., 2024b; Xiong et al., 2024; Lyu et al., 2025; Es et al., 2024).

In this work, we propose Multi-Head RAG (MRAG): a scheme that addresses the above problem (**contribution 1**). Common practice in many modern RAG designs is to use embeddings derived from *last-layer decoder block activations* of a decoder-based *embedding LLM* (a language model specifically fine-tuned to provide high-quality embeddings). Examples of such embedding models are `SFR-Embedding-Mistral` and `E5-Mistral-7B`. Our key idea is to extend this design by incorporating activations from the *multi-head attention (MHA) modules of decoder blocks* as embedding sources. This enables the representation of multiple distinct aspects of the input text. Specifically, a Transformer consists of a stack of blocks (e.g., 96 in GPT-3 (Wang et al., 2025b)), each containing an MHA module with multiple *heads* that are trained with separate parameter sets. Through a survey of literature into Transformer design and interpretability, we find empirical and theoretical support for the conjecture that *different heads specialize in different aspects of the input*

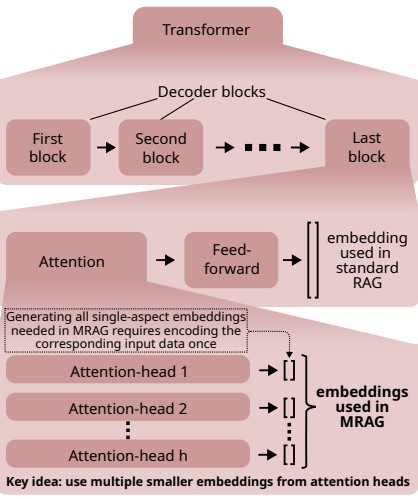

Figure 1: An overview of the decoder architecture, and a comparison of how standard RAG and Multi-Head RAG embeddings are generated.

(details are in Section 2.4). This enables efficient *multi-aspect embeddings* without increasing space or compute costs compared to standard RAG, and without requiring *any* additional fine-tuning or architectural modifications to the base model.

Considering multi-aspectuality comes with challenges. For example, it is unclear how to assess whether a RAG solution does indeed harness multiple aspects when fetching documents. For this, we develop a multi-aspect RAG pipeline that includes data preparation and query processing with both multi-aspect retrieval and ranking schemes (**contribution 2**). We also establish an evaluation methodology and provide multi-aspect datasets, complementing existing RAG benchmarks (Chen et al., 2024b) (**contribution 3**). We ensure the relevance of our RAG datasets in real use cases by working directly with tech leaders (e.g., a generative AI division head) from 3 corporations, all of which actively use RAG in their own LLM infrastructures. We illustrate the advantages of MRAG over 18 traditional and modern RAG designs in various design criteria and in both time and space complexities (**contribution 4**). In evaluation, MRAG enhances the relevance of retrieved documents by up to 20% over modern RAG baselines, offers comparable performance without degradation for single-aspect queries, and benefits the downstream LLM generation (**contribution 5**). Thanks to its simplicity, MRAG can be seamlessly integrated into any stores while its benchmarking methodology can straightforwardly extend benchmarks such as RAGAs (**contribution 6**).

## 2  MRAG: DESIGN & IMPLEMENTATION

A typical RAG scheme (see Figure 2) consists of two main parts: **data preprocessing Ⓐ** and **query execution Ⓑ**; both parts heavily use an **embedding model Ⓒ** and the **data store Ⓓ**. During preprocessing, each document in the database is encoded into one or more embeddings using the embedding model; these embeddings are stored with that document (usually as a key-value pair in a vector DB). During query execution, the embedding of the user-provided query is constructed using the same embedding model; then, the *retriever* fetches candidate documents based on embedding similarity, the *reranker* refines their ordering using more precise scoring, and the *reader* uses a downstream generative model to synthesize the final answer from the top-ranked documents.

### 2.1  CONSTRUCTING MULTI-ASPECT EMBEDDINGS []

An embedding is constructed for each data item in a pre-existing database (part Ⓐ) and for the user query (part Ⓑ). In standard modern RAG, given an input chunk of $n$ tokens constituting a document

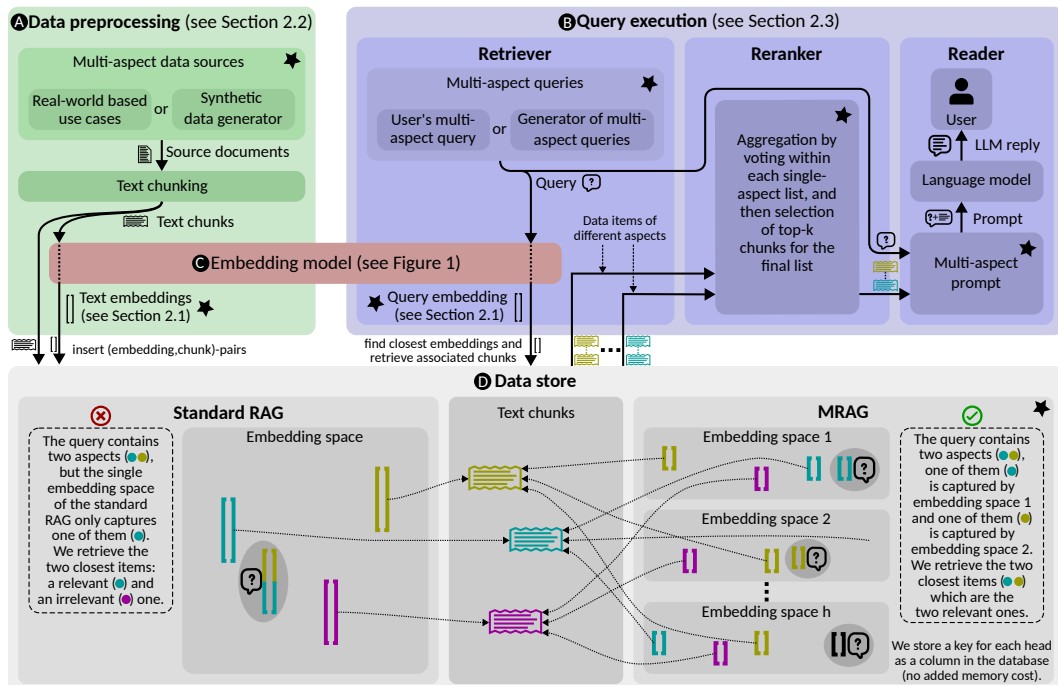

Figure 2: Overview of the MRAG pipeline, consisting of two parts: data preparation **A** and query execution **B**. The embedding model **C** and the data store **D** are used by both parts. The data store **D** contains text embeddings [] linking to text chunks ≣ reflecting three different aspects (cyan, magenta, yellow). Blocks marked by a star ✦ are a novelty of this work.

or chunk to be embedded, one obtains a corresponding embedding by applying the embedding model to the data item and extracting the activation of the final feed-forward layer for the last token $\mathbf{x}_n$, i.e., `feed-forward(multi-head($\mathbf{x}_n$))` $\in \mathbb{R}^d$. Appendix B provides additional mathematical details.

In MRAG, instead of relying on this single embedding, the activations of all $h$ attention heads are obtained *before* they are merged by the final projection layer. Specifically, for the last token $\mathbf{x}_n$, we extract the set of head-specific vectors $\mathcal{S} = \{\mathbf{e}_k\}_{k=1}^h$, where each $\mathbf{e}_k = \text{head}^k(\mathbf{x}_n) \in \mathbb{R}^{d/h}$ is referred to as a "*single-aspect embedding*" and $\mathcal{S}$ is called a "*multi-aspect embedding*". This provides $h$ semantically diverse embeddings per input, reflecting the different perspectives captured by each attention head. Since these vectors are extracted from the same internal activation used by standard RAG, the overall space and time requirements remain unchanged.

## 2.2 DATA PREPROCESSING **A**

We populate a data store **D** with multi-aspect embeddings [] for their corresponding documents 📄 or text chunks ≣. Unlike in standard RAG, where a single embedding [] points to a single text chunk ≣, in MRAG, each out of $h$ *single-aspect* embeddings [] points to the original text chunk ≣ (i.e., the data store **D** contains $h$ embedding spaces). This crucial feature allows MRAG to compare query ⑦ and text chunks ≣ in multiple embedding spaces that capture multiple aspects of the data.

MRAG performs one-time *importance scoring* for each embedding space $i$, capturing its relevance to the dataset. Each score $s_i$ combines: (i) the average L2 norm $a_i$ of embeddings in space $i$, reflecting *attention strength*, and (ii) the average cosine distance $b_i$ between sampled embeddings, approximating its *semantic spread*. The final score is $s_i = a_i \cdot b_i$, encouraging both head relevance (through high $a_i$) and high representational diversity (through high $b_i$). Full details are provided in Algorithm 1, Appendix C.1.

## 2.3 QUERY EXECUTION

During query execution, MRAG first generates a multi-aspect embedding $\mathcal{S}$ of the input query (cf. Section 2.1). These embeddings are computed fully in parallel during the same inference pass, so this multi-vector representation introduces no additional latency.

**Retriever.** Given $\mathcal{S}$, the retriever conducts parallel retrieval of the top-$c$ nearest chunks within each embedding space, and the aggregation of the resulting candidates across all $h$ spaces.

**Reranker.** Once $hc$ candidate chunks are retrieved across all heads, the ranker stage consolidates them into a single list of top-$k$ results using a simple but effective voting strategy. For each candidate chunk at position $p$ in the ranked list for space $i$, we assign a score of $s_i \cdot 2^{-p}$, where $s_i$ is the precomputed importance score. This exponentially discounts lower-ranked candidates and balances influence across heads. The final top-$k$ list is obtained by globally sorting all candidates by these scores. The voting procedure is described in Algorithm 2 in Appendix C.2.

**Reader.** The top-$k$ retrieved results are inserted into the LLM context using a multi-aspect prompt template (prompts are fully specified in Appendix D). Each result is placed in a separate section of the prompt. Stored metadata can be included alongside each chunk to provide additional context, such as the aspect or the chunk category.

## 2.4 Capturing Multi-Aspectuality Without Additional Training

A key design decision in MRAG is to use the hidden representations immediately after the attention block in the last decoder layer, *without additional fine-tuning*, to avoid training overhead and make deployment easy. This is motivated by growing evidence that attention heads in Transformer models naturally converge during training to focus on distinct aspects of the input. For example, Wang et al. (2025a) show that heads diverge into clusters specialized for different input patterns, while Olsson et al. (2022) and others (McDougall et al., 2024; Gould et al., 2024) discover that heads focus on repeated sequences or named entities. Similar trends are observed in BERT (Clark et al., 2019; Kovaleva et al., 2019; Htut et al., 2019). We provide more details in a brief literature survey (Appendix E.1). Given these findings, we assume that even without additional fine-tuning, the embeddings output by MHA already encode multi-aspectuality suitable for downstream retrieval. Our own brief attention pattern analysis (Appendix E.2) confirm shifting token focus across heads.

## 2.5 Parallelization and Systems Considerations

MRAG evenly distributes the embedding dimensionality across attention heads, keeping total storage and compute $O(nd)$ (same as standard RAG; we show a more detailed complexity analysis in the following section). It uses off-the-shelf ANN indexes that support parallel subspace search, and it avoids dynamic or variable-length vectors that complicate indexing and caching.

Modern vector databases already support the parallel multi-vector search required by our approach. Milvus can execute distributed, parallel ANN queries across shards with low latency (Wang et al., 2025c; Clavié et al., 2024; Wang et al., 2021). Pinecone provides cascading retrieval with fixed-size multi-vector encodings (such as ConstBERT) to maintain predictable costs. Furthermore, ESPN tackles multi-vector retrieval at SSD and GPU levels by implementing prefetching pipelines and storage bypass mechanisms, achieving near-memory query latency, even when dealing with large indices (Shrestha et al., 2024). Taken together, these technologies illustrate that multi-vector retrieval, especially when using fixed-size vectors as MRAG does, can be implemented at scale with negligible practical overhead.

## 3 Compute & Storage Complexity Analysis

We analyze the runtime and space complexity of MRAG and 18 RAG baselines, the results are in Table 1 (the runtime metrics of work/depth and the notation are explained in the table caption). Derivation details and a broader discussion are in Appendix F. Overall, MRAG achieves competitive results in all aspects. At inference time, it extracts $h$ attention-head embeddings in parallel from a single forward pass, resulting in the same latency as standard RAG. Preprocessing is lightweight, requiring only one pass per document and simple statistics for head scoring, avoiding the cost of training or complex structure construction. Storage overhead is minimal, as $h$ single-aspect embeddings per data item have the same dimensionality as a standard embedding. In contrast, prior schemes like Poly-encoder (Humeau et al., 2020) and ColBERT (Khattab & Zaharia, 2020) incur significant cost due to using many token-level embeddings or time-consuming training rounds.

## 4 Benchmarking Multi-Aspectuality

To assess how well MRAG performs on multi-aspect queries, we need (1) datasets of multi-aspectual documents, (2) queries to the LLM that require retrieving multi-aspect documents, and (3) metrics that assess how well a RAG scheme retrieves such multi-aspect data. We now summarize these three elements; details are in Appendix G.

Table 1: **Time and storage complexity of different RAG schemes; MRAG is the only scheme to match the results of a plain vanilla RAG**. **Work** is the total number of all operations, **Depth** is runtime complexity assuming unlimited parallel processing units; both are established measures of analyzing parallel algorithms. **Retrieval** describes steps during inference, including any postprocessing before returning documents or summaries. **Preprocessing** includes all steps before inference. $n$: number of data items in a database (i.e., document chunks), $d$: embedding dimensionality, $k$: number of retrieved top chunks per single user query, $l_q$: average token length of the query, $l_d$: average token length of the document, $W_m/D_m$: work/depth to run a transformer-based model, $W_e/D_e$: work/depth to embed a graph, $W_i/D_i$: work/depth to index a graph, $s$: polynomial function that models the complexity of various additional scheme-specific design decisions, heuristics, etc., which cannot be straightforwardly modeled with closed-form expressions (e.g., count of graph communities, toy-graph size, self-note length, BM25 matching cost, keyword matching cost); it scales at most linearly with $n$ and is typically considerably larger than $O(1)$.

| Scheme | Retrieval | | Preprocessing | | Storage |
|---|---|---|---|---|---|
| | Work | Depth | Work | Depth | |
| **Vanilla RAG** | $O(W_m + nd)$ | $O(D_m + \log d)$ | $O(W_m n)$ | $O(D_m)$ | $O(nd)$ |
| Poly-encoders (Humeau et al., 2020) | $O(W_m + nd + s(n)d)$ | $O(D_m + \log d)$ | $O(W_m n)$ | $O(D_m)$ | $O(nd)$ |
| Lewis et al. (2020) | $O(2W_m + nd)$ | $O(2D_m + \log d)$ | $O(W_m n)$ | $O(D_m)$ | $O(nd)$ |
| ColBERT (Khattab & Zaharia, 2020) | $O(W_m + nd l_q l_d)$ | $O(D_m + \log d l_d)$ | $O(W_m n)$ | $O(D_m)$ | $O(nd l_d)$ |
| EMDR (Singh et al., 2021) | $O((k+1)W_m + nd)$ | $O(2D_m + \log d)$ | $O(W_m n)$ | $O(D_m)$ | $O(nd)$ |
| Self-RAG (Asai et al., 2024) | $O((2k+1)W_m + nd + k)$ | $O(2D_m + \log d)$ | $O(W_m n)$ | $O(D_m)$ | $O(nd)$ |
| Chain-of-Note (Yu et al., 2024) | $O((ks(n)+1)W_m + nd)$ | $O((ks(n)+1)D_m + \log d)$ | $O(W_m n)$ | $O(D_m)$ | $O(nd)$ |
| RAPTOR (Sarthi et al., 2024) | $O(W_m + nd + s(n)d)$ | $O(D_m + \log d)$ | $O(W_m n + W_m s(n))$ | $O(D_m)$ | $O(nd + s(n)d)$ |
| RAGraph (Jiang et al., 2024) | $O(W_m + nd + s(n)d)$ | $O(D_m + \log d)$ | $O(W_m + W_e)$ | $O(D_m + D_e)$ | $O(nd + s(n)d)$ |
| RQ-RAG (Chan et al., 2024) | $O(s(n)(W_m + nd))$ | $O(s(n)(D_m + \log d))$ | $O(W_m n)$ | $O(D_m)$ | $O(nd)$ |
| ActiveRAG (Xu et al., 2024) | $O(3W_m + nd)$ | $O(2D_m + \log d)$ | $O(W_m n)$ | $O(D_m)$ | $O(nd)$ |
| HiQA (Chen et al., 2024c) | $O(W_m + nd + s(n)d)$ | $O(D_m + \log d)$ | $O(W_m n + s(n)d)$ | $O(D_m)$ | $O(nd + s(n)d)$ |
| GraphRAG (Edge et al., 2025) | $O(s(n)(W_m + d))$ | $O(D_m + s(n)d)$ | $O(W_m n + W_m s(n) + W_e)$ | $O(D_m + D_e)$ | $O(nd + s(n)d)$ |
| Fusion RAG (Rackauckas, 2024) | $O(s(n)W_m + knd)$ | $O(2D_m + \log d)$ | $O(W_m n)$ | $O(D_m)$ | $O(nd)$ |
| Meta-chunking (Zhao et al., 2025b) | $O(W_m + (n + s(n))d)$ | $O(D_m + \log d)$ | $O(l_d W_m + s(n)W_m)$ | $O((l_d + 1)D_m)$ | $O(nd + s(n)d)$ |
| MoC (Zhao et al., 2025a) | $O(W_m + (n + s(n))d)$ | $O(D_m + \log d)$ | $O(W_m n + s(n))$ | $O(D_m)$ | $O(nd + s(n)d)$ |
| Parametric RAG (Su et al., 2025) | $O(W_m + nd + s(n))$ | $O(D_m + \log d + s(n))$ | $O(W_m n s(n))$ | $O(D_m s(n))$ | $O(ns(n))$ |
| SuperRAG (Yang et al., 2025) | $O(W_m + nd + W_i)$ | $O(D_m + \log d + D_i)$ | $O(W_m n + s(n) + W_e)$ | $O(D_m + D_e)$ | $O(nd + s(n))$ |
| HiRAG (Huang et al., 2025a) | $O(W_m + nd + W_i)$ | $O(D_m + \log d + D_i)$ | $O(W_m n + s(n) + W_e)$ | $O(D_m + D_e)$ | $O(nd + s(n))$ |
| **MRAG [This Work]** | $O(W_m + nd)$ | $O(D_m + \log d)$ | $O(W_m n)$ | $O(D_m)$ | $O(nd)$ |

We construct three **multi-aspect datasets**: (1) a synthetic Wikipedia dataset with documents sampled from clearly distinct categories (e.g., countries, shipwrecks, board games); (2) a real-world-based legal document dataset annotated by a legal areas (energy law, family law, criminal law, etc.) or document language style (aggressive, mild, neutral, etc.); and (3) a real-world-based dataset of industry accident reports, categorized by cause. For each dataset, we generate **multi-aspect queries** using GPT-4o, combining $n$ distinct categories into single queries (with $n \in \{1, 2, 3, 4, 5, 6, 10, 15, 20, 25\}$). For example, a query with 10 aspects must contain a question about 10 different documents from 10 different categories.

To evaluate retrieval performance, we introduce three **metrics for assessing multi-aspectuality**. Let $Q$ be a query, $S$ a Reranker scheme, and $Q_{rel}$ the ideal set of $n$ relevant documents. The *Retrieval Success Ratio* is defined as $\Xi(Q, n) = \frac{|S(Q,n) \cap Q_{rel}|}{|Q_{rel}|}$, measuring the fraction of exactly matched documents. The *Category Retrieval Success Ratio* $\Xi_c$ extends this by also accepting matches from the same categories as those in $Q_{rel}$ even if the exact document has not been matched. Finally, we define a tunable *Weighted Success Ratio* $\Xi_w = \frac{w \cdot \Xi + \Xi_c}{w+1}$, allowing the user to adjust the trade-off between exact and category-level matches via the weight $w$.

We motivate category-level retrieval using both previous works (V et al., 2025; Liu et al., 2021; Chen et al., 2024a; Xiao et al., 2024) as well as our hands-on experience in legal and industrial accident analysis. Namely, even when exact matches are missing, retrieving documents from the same semantic category can enhance generation, as supported by the classic *clustering hypothesis* in information retrieval (V et al., 2025; Liu et al., 2021): documents in the same "neighborhood" are likely to be relevant to the same query. Recent studies further validate this across open-domain QA and ontology-guided RAG (Chen et al., 2024a; Xiao et al., 2024).

## 5 EVALUATION

We now illustrate the advantages of MRAG over the state of the art. Further details on evaluation setup and additional results are in – respectively – Appendix H and I.

**Comparison Baselines.** We consider three main baselines: **Standard RAG**, **Split RAG**, and **Fusion RAG** (Rackauckas, 2024). The first represents a modern RAG pipeline in which each document uses the activations of the last decoder layer as its embedding. The second is a blend between Standard RAG and MRAG; it splits the activation of the last decoder layer in the same way as MRAG and applies a voting strategy. The purpose of Split RAG is to show that *MRAG's benefits come from using the multi-head output as embedding and **not** merely using multiple embedding spaces*. Additionally, we consider **Fusion RAG** (Rackauckas, 2024), an optional mechanism that we harness to *further enhance the benefits of MRAG at the tradeoff of additional tokens* (detailed in Section 5.3).

**Embeddings & models.** While MRAG allows for extracting multi-aspect embeddings from *any* block, we found that the last MHA works best in our experiments. MRAG can use any embedding

model with MHA; we consider two embedding models from the MTEB leaderboard (Huggingface, 2025), the SFR-Embedding-Model (Meng et al., 2024) and the e5-mistral-7b-instruct (Wang et al., 2024), both based on the Mistral 7B architecture with 32 decoder blocks and 32 attention heads.

## 5.1 MRAG Delivers Superior Performance for Multi-Aspect Queries

We start with the **Wikipedia multi-aspect dataset** and corresponding queries (cf. Section 4). In each query, we mention the documents to be fetched in the text and then assess the success ratio of different RAG strategies in finding these documents and their categories (a full example of such a query is in Figure 10 in Appendix G). We show first the absolute retrieval performance of MRAG over Standard RAG in Figure 3. We fix the number of aspects present in the queries to 10, and vary the total number of retrieved documents from 10 to 30. MRAG consistently outperforms Standard RAG ($> 10\%$ increase in the retrieval success ratio on average for exact document matches). Moreover, the retrieval performance increase is even more significant on category matches ($> 25\%$ increase in the retrieval success ratio on average). The performance increase is further detailed in the histograms on the right side. Here, for a specific number of documents fetched, MRAG's histogram indicates a better distribution of retrieval success ratios (across all 25 queries). This gain stems from MRAG's ability to decompose query semantics across multiple attention heads, increasing the likelihood of matching each aspect with a semantically aligned document.

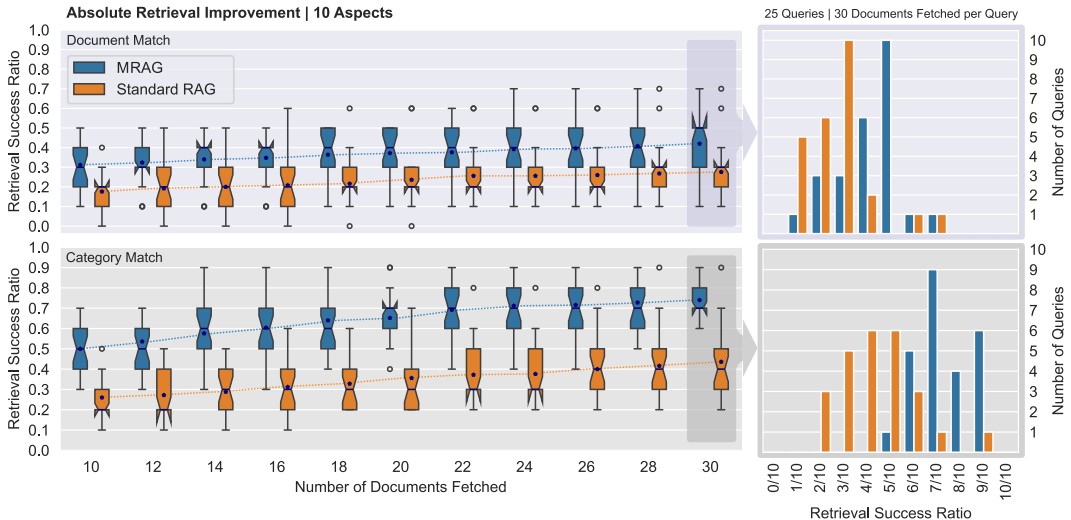

Figure 3: **Retrieval success ratio over 25 queries between MRAG and Standard RAG**; each query uses 10 different aspects. The top part presents **exact document matches**, the bottom part presents **category only matches** (we explain the metrics in Sec. 4). A histogram is presented for a specific sample to showcase the detailed distribution among the 25 queries (there are 30 documents fetched for *each* query).

Next, Figure 4 shows the relative weighted performance improvement of MRAG with respect to Standard RAG as we vary the number of aspects present in the queries. We show data for two different embedding models (SFR and e5). MRAG consistently outperforms the Standard RAG by 10-20% on average, not only across the number of documents fetched, but also across the increasing counts of aspects present in the replies, and does it for both embedding models. This robustness suggests that MRAG scales better than Standard RAG with query complexity, as its multi-head representation distributes semantic load more evenly than the single-vector baseline.

To further illustrate advantages of MRAG, we also consider two **real-word use cases** from in-house industry data analytics projects, namely, the synthesis of legal documents and the analysis of causes of chemical plant accidents. The results are in Figure 5. In the former (the left side), the task is to create a document based on user requirements that may be related to different *aspects*, for example to the law being considered (e.g., the British or the US one), the subject (e.g., energetic or civil), the style of the document (e.g., aggressive or mild), and others. This task is executed with RAG that can fetch documents from a database. In the latter (the right side), the task is to discover a cause of an accident. Here, one also wants to retrieve documents from a database that should be used in the LLM context to facilitate discovering the cause of the accident. The causes are grouped in categories such as utility impact due to severe weather, lack of preparedness and planning, incorrect installation

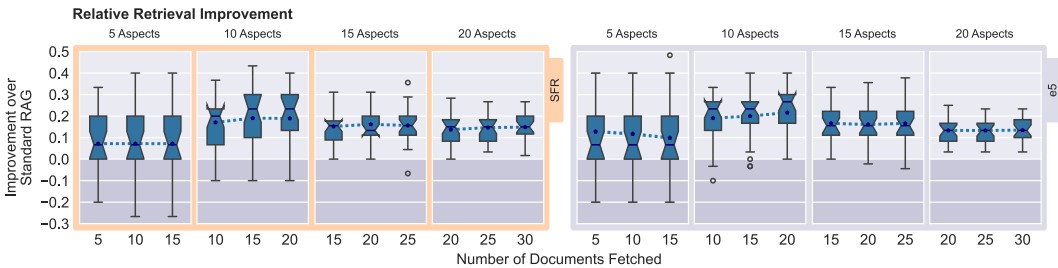

Figure 4: **Relative retrieval improvement of MRAG over Standard RAG** across queries with different numbers of aspects and different embedding models (SFR in the left side, e5 in the right side).

of equipment, lack of maintenance, et cetera. Similarly to the previous analyses, we measure the retrieval success ratio over corresponding databases. MRAG offers advantages over other schemes.

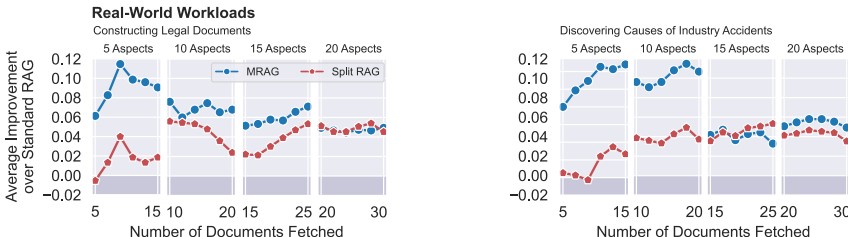

Figure 5: **Average improvement of the retrieval success ratio of MRAG and Split RAG over Standard RAG** for two real-world workloads *constructing legal documents* (left) and *discovering causes of industry accidents* (right).

We also **delve deeper into the underlying factors for MRAG's performance gains**. For this, we compare MRAG to the Split RAG baseline in Figure 6. The blue plots show the relative weighted performance of MRAG and Split RAG over Standard RAG. MRAG performs better than Split RAG, illustrating that its *high accuracy is due to the actual multi-head part*, and not merely just partitioning the vector and using multiple embedding spaces.

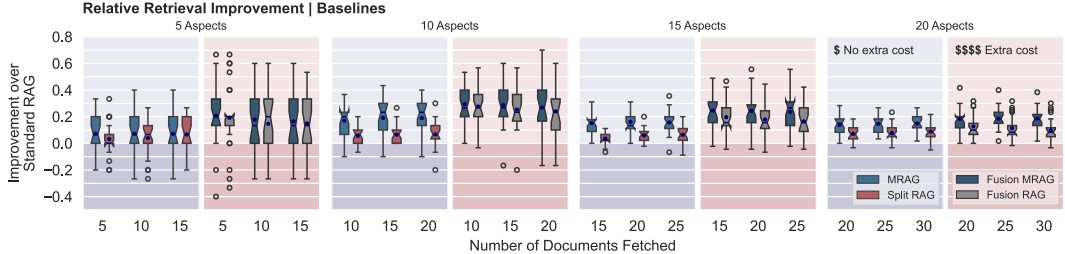

Figure 6: **Relative retrieval improvements of MRAG over Standard RAG** for the SFR embedding model compared with **Split RAG** (the blue plots), and the **relative retrieval improvements of Fusion MRAG over Standard RAG** compared with **Fusion RAG** (the red plots).

### 5.2 MRAG ENSURES HIGH PERFORMANCE FOR SINGLE-ASPECT QUERIES AS WELL

We additionally show in Table 2 that MRAG performs on-par with Standard RAG on queries where only a single aspect is expected. This confirms that MRAG's design generalizes well: in low-aspect settings, the aggregation over heads still captures dominant semantics, effectively collapsing into a strong single-vector representation with only a negligible accuracy drop for single-aspect tasks.

### 5.3 MRAG SEAMLESSLY ENHANCES EXISTING RAG SCHEMES

MRAG's simplicity ensures that it can be seamlessly integrated with other RAG approaches. As an example, we combine MRAG with *Fusion RAG*, which uses an LLM (additional token cost) for even more accurate retrieval. Fusion RAG uses an LLM to create a fixed number of questions about the RAG query. Each question is separately applied through an embedding model using Standard RAG. This relies on multiple LLM calls and heuristic reranking, inflating latency and cost.

We apply MRAG to each of these questions and denote the combined scheme as *Fusion MRAG*. Red plots of Figure 6 show that Fusion MRAG consistently outperforms pure Fusion RAG, indicating

Table 2: **Retrieval success ratio** (the exact document match) for 25 **single-aspect** queries on different datasets (Multi-Aspect Dataset, Legal Dataset, Accidents Dataset), using different embedding models (SFR, e5). For every query, a specific single document (single-aspect) is expected to be among the fetched documents for the retrieval to be classified as successful.

| Documents Fetched | Wikipedia Dataset | | | | | Legal Dataset | | | Accidents Dataset | |
| | SFR | | — | e5 | | SFR | | — | SFR | |
| | MRAG | Standard RAG | — | MRAG | Standard RAG | — | MRAG | Standard RAG | — | MRAG | Standard RAG |
| --- | --- | --- | --- | --- | --- | --- | --- | --- | --- | --- | --- |
| 1 | 24/25 | 25/25 | | 24/25 | 25/25 | | 24/25 | 24/25 | | 25/25 | 25/25 |
| 2 | 25/25 | 25/25 | | 25/25 | 25/25 | | 25/25 | 25/25 | | 25/25 | 25/25 |
| 3 | 25/25 | 25/25 | | 25/25 | 25/25 | | 25/25 | 25/25 | | 25/25 | 25/25 |

that these optimizations can be combined together. However, both Fusion strategies introduce a greater variance than MRAG and additional costs in terms of compute, latency, and tokens.

## 5.4 MRAG Seamlessly Enhances Downstream LLM Generation

We also illustrate that MRAG enhances the downstream LLM generation with its improved multi-aspect retrieval. To show this, we use a sampled subset of multi-aspect Wikipedia queries, for which we applied both Standard RAG and MRAG to retrieve supporting documents. These documents are then integrated into the prompt templates and are passed to the LLM to answer the original query with the aid of the retrieved data. To quantify the effect from RAG, we count facts in the LLM output (e.g., years or named entities such as locations). On average, MRAG generations contain on average 15.4 pieces of factual information per query, compared to 11.2 for Standard RAG. The average improvement further confirms MRAG's advantages. By explicitly leveraging multi-aspectual embeddings from MHA, one increases the likelihood that the LLM generation includes diverse and complementary facts, especially for complex queries spanning multiple domains (e.g., combining historical events with technological timelines). Overall, MRAG helps the LLM in delivering richer, more comprehensive responses.

## 5.5 MRAG Ensures No Latency and Storage Overheads

MRAG introduces no latency overhead at query embedding time, as all head-level embeddings are extracted in parallel from a single standard forward pass. Retrieval across embedding subspaces is also fully parallelizable with modern vector databases (Pan et al., 2024; Ma et al., 2025), and our use of a modest number of heads (e.g., 16–32 for SFR-Embedding-Model) ensures this parallelism is within easy reach. Moreover, as the total dimensionality of embeddings remains unchanged (e.g., 1024 split across 32 heads), MRAG needs no additional storage compared to standard RAG.

## 5.6 Further Analyses & Ablation Studies

Additional analyses are in Appendix I, they include analyzing the impact of using embeddings from **different decoder blocks** (rather than the last one), **scalability of preprocessing**, and **additional voting strategies for reranking**. These analyses all confirm the previous findings of MRAG broadly outperforming other baselines.

## 6 Related Work & Advantages of Multi-Head RAG

**RAG Solutions.** We compare MRAG to a large number of both traditional and modern RAG solutions in Table 3 in terms of design advantages (complexity analysis and empirical evaluation are in, respectively, Sections 3 and 5). While prior RAG systems support retrieving multiple documents for a single query (e.g., RAG (Lewis et al., 2020), EMDR (Singh et al., 2021)), none of them generates *multi-aspect* embeddings per document, and therefore do not offer multi-aspectuality. Similarly, although these systems employ Transformer architectures with MHA, they make no use of the MHA internal structure. Other methods such as Poly-encoder (Humeau et al., 2020) and ColBERT (Khattab & Zaharia, 2020) do produce multiple embeddings per document, but for a different reason: they are based on models such as BERT, which inherently yields token-level embeddings. Thus, these models require multiple vectors per document simply to cover its content at the token level, not to represent distinct semantic aspects. In contrast, MRAG leverages the powerful internal structure of modern decoder-based LLMs, where a small number of vectors derived from MHA (or even a single vector from the final decoder block, as in standard RAG) suffices to represent the semantics of an entire document or a chunk, as indicated by recent work (Lee et al., 2025; Besta et al., 2025).

Building on this foundation, MRAG achieves further practical advantages. It introduces *scalable preprocessing* since (1) global embeddings are computed in a single forward pass per document and (2) scoring of heads is straightforwardly parallelizable. At inference time, it incurs *no additional cost* relative to standard RAG – in contrast to token-level approaches, which require computing nu-

Table 3: **Comparison of the advantages of different RAG schemes** (sorted top to bottom chronologically). **No additional training**: does a given scheme require additional training beyond standard pre-training and fine-tuning? **Works with any MHA LLM**: does a given scheme can seamlessly work with any multi-head attention (MHA) LLM? **Extensibility to other modalities**: could a given scheme be relatively easily used beyond LLMs, e.g., with Graph Foundation Models (GFMs), Vision Models, and others? **No overhead at inference**: does a given scheme enable zero additional overhead at the inference? **Scalable preprocessing, low storage overhead**: does a given scheme enable scalable preprocessing and little storage overhead, respectively? (details in Section 3 and Table 1). **Multi-Aspectuality**: does a given scheme enable extracting multiple aspects of indexed data? "■": full support, "◧": partial support, "✗": no support, "?": unknown.

| Scheme | No additional training | Works with any MHA LLM | Extensibility to other modalities | No overhead at inference | Scalable preprocessing | Low storage overhead | Multi-Aspectuality |
|---|---|---|---|---|---|---|---|
| Poly-encoders (Humeau et al., 2020) | ✗ | ✗ (BERT based) | ✗ | ✗ | ■ | ◧ | ✗ |
| Lewis et al. (2020) | ✗ | ■ | ✗ | ✗ | ■ | ◧ | ✗ |
| ColBERT (Khattab & Zaharia, 2020) | ✗ | ✗ (BERT based) | ✗ | ◧ | ■ | ◧ | ◧ |
| EMDR (Singh et al., 2021) | ✗ | ■ | ✗ | ■ | ◧ | ◧ | ✗ |
| Self-RAG (Asai et al., 2024) | ✗ | ■ | ✗ | ✗ | ✗ | ✗ | ✗ |
| Chain-of-Note (Yu et al., 2024) | ✗ | ◧ | ✗ | ✗ | ■ | ◧ | ✗ |
| RAPTOR (Sarthi et al., 2024) | ■ | ■ | ✗ | ✗ | ■ | ◧ | ✗ |
| RAGraph (Jiang et al., 2024) | ■ | ✗ (only GFMs) | ✗ (only GFMs) | ◧ | ■ | ✗ | ◧ |
| RQ-RAG (Chan et al., 2024) | ✗ | ■ | ✗ | ✗ | ◧ | ■ | ✗ |
| ActiveRAG (Xu et al., 2024) | ■ | ■ | ✗ | ✗ | ◧ | ◧ | ✗ |
| HiQA (Chen et al., 2024c) | ■ | ■ | ✗ | ✗ | ◧ | ✗ | ◧ |
| GraphRAG (Edge et al., 2025) | ■ | ■ | ? | ✗ | ■ | ■ | ◧ |
| Fusion RAG (Rackauckas, 2024) | ■ | ◧ | ✗ | ✗ | ■ | ◧ | ✗ |
| Meta-chunking (Zhao et al., 2025b) | ■ | ◧ | ✗ | ■ | ■ | ◧ | ✗ |
| MoC (Zhao et al., 2025a) | ✗ | ◧ | ✗ | ✗ | ■ | ■ | ✗ |
| Parametric RAG (Su et al., 2025) | ✗ | ◧ | ✗ | ✗ | ◧ | ✗ | ✗ |
| SuperRAG (Yang et al., 2025) | ■ | ■ | ■ | ✗ | ◧ | ◧ | ✗ |
| HiRAG (Huang et al., 2025a) | ■ | ■ | ? | ✗ | ■ | ◧ | ✗ |
| **MRAG [This work]** | ■ | ■ | ■ | ■ | ■ | ■ | ■ |

merous pairwise token similarities during retrieval, our method only compares compact, chunk-level vectors. *Storage overhead is also minimal*: the MHA-derived embeddings match the dimensionality of standard last-layer embeddings (detailed time and storage complexity analyses are in Section 3). Finally, MRAG requires *no training* and is *highly versatile*: it can be applied to any Transformer model with MHA, and is easily extensible to other modalities such as vision and graph data, where Transformer architectures are now common.

**Reranking.** Retrieval is sometimes enhanced by a **reranking** phase (Rosa et al., 2022; Nogueira & Cho, 2020; Nogueira et al., 2020; Li et al., 2023; Gao et al., 2021; MacAvaney et al., 2019). Here, after retrieving a set of relevant chunks, they are re-ranked using specialized models. *In this work, we provide a heuristic reranker that considers multi-aspectuality, but we design MRAG specifically so that it can be seamlessly used in conjunction with any other existing cross-encoders.*

**Multi-Head Embeddings Outside RAG.** Several methods propose a new model variant or architectural modification to the Transformer in order to better exploit MHA (Park et al., 2020; Huang et al., 2019; Wang et al., 2020; Xue & Aletras, 2023). For example, MHSAN (Park et al., 2020) introduces a novel visual-semantic embedding network that extracts multiple region- and phrase-sensitive features using MHA. Wang et al. (2020) develop a speaker embedding network that explicitly enforces head-level diversity via contrastive learning across resolutions. Xue & Aletras (2023) propose PIT, a Transformer variant that composes attention heads across layers to reduce redundancy and enable efficient inference. Vashisht et al. (2025) introduce MAGE, a technique that mixes heads across models to improve generalization through stochastic head combinations. *In contrast, MRAG is the first method to harness embeddings from MHA in pre-trained, decoder-based embedding LLMs for the purpose of more effective RAG. Unlike the above works, MRAG requires no architectural modifications, no training, and no custom modules, being fully plug-and-play.*

## 7 CONCLUSION

We introduced Multi-Head RAG (MRAG), a simple yet powerful extension to RAG that leverages the multi-head attention (MHA) activations of decoder models to capture multiple semantic aspects of a query or document. Motivated by the challenge of retrieving semantically diverse documents for multi-aspect queries, which is a common need in real-world applications, MRAG generates a set of aspect-specific embeddings without requiring extra training, model calls, or increased storage. Through a comprehensive evaluation including synthetic and industrial datasets, and tasks ranging from the accuracy of retrieval to the quality of downstream LLM generations, we demonstrate that MRAG consistently improves retrieval relevance (up to 20%) and yields richer, more factual outputs. Unlike many advanced RAG baselines, MRAG remains plug-and-play, scalable, and efficient—making it a practical and principled solution for high-precision multi-aspect retrieval in LLM-driven systems.

**Ethics statement.** This work uses datasets and models that are publicly available or licensed for use. We did not conduct research involving human subjects and did not collect or share any personal or sensitive information. All data in our case studies was de-identified and used according to its license terms. Our work focuses on retrieval and evaluation methods for multi-aspect queries and does not create new risks.

**Reproducibility statement.** We make our results easy to reproduce. We release code, configuration files, and scripts to rebuild indexes, run retrieval/reranking, and execute all evaluations, along with detailed instructions, hyperparameters, prompts, and random seeds. We document software and hardware versions and model/dataset identifiers. Where licensing prevents sharing specific corpora, we provide preprocessing scripts and exact instructions to reconstruct them. All figures and tables in the paper can be regenerated.

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

APPENDIX

## A REVIEW OF MULTI-ASPECTUALITY IN INDUSTRY

Real-world decision-making and analysis tasks often require a *holistic integration of multiple, semantically distinct information sources*. Literature from diverse domains—from safety investigations to diagnostics and enterprise analytics—consistently emphasizes that no single data stream suffices for complex tasks. Instead, success comes from combining heterogeneous elements (design details, human factors, environmental context, etc.) into a unified understanding. Below we highlight several examples that support this multi-aspectual reasoning framework.

In **safety-critical environments**, accident and incident investigations rely on aggregating information from disparate sources. For example, a U.S. Federal Railroad Administration report stresses that collecting multiple sources of information is essential for effective event reconstruction (Office of Railroad Safety, 2014). The Columbia Accident Investigation Board similarly combined telemetry, debris forensics, environmental data, and even autopsy reports to reconstruct the chain of events behind the Space Shuttle Columbia disaster (Columbia Accident Investigation Board, 2003). These investigations exemplify the need to integrate technical, procedural, and human-centered aspects of data to obtain actionable conclusions.

In **medicine**, particularly oncology, the concept of *integrated diagnostics* exemplifies multi-aspect decision-making. Physicians routinely combine patient histories, radiological scans, lab values, pathology slides, and genetic tests to form a diagnosis. This is no longer optional: as modern datasets become more complex, integrating semantically distinct modalities has become necessary to reach accurate, personalized outcomes (Messiou et al., 2023; Kalia, 2013).

The same applies to **enterprise settings**. For example, airport operations management integrates foot traffic sensors, weather feeds, and gate schedules – among others – to optimize personnel allocation and prevent congestion (Zografos et al., 2013). Similarly, in legal and financial firms, cross-silo systems integrate internal metrics (e.g., billing and staffing) with external sources (e.g., news, contracts, social media) to guide decision-making and strategic planning (OpenText, 2022). These examples show that modern organizational workflows demand the integration of multiple semantically distinct data sources.

In summary, across domains like industrial safety, healthcare, and business intelligence, it is widely recognized that *multi-aspectuality*—combining diverse, independently relevant information fragments—is essential for accurate and effective decision-making (Office of Railroad Safety, 2014; Columbia Accident Investigation Board, 2003; Reason et al., 2006; Kalia, 2013; Packham, 2017; Messiou et al., 2023; Multer et al., 2013; Coury et al., 2010; Harle, 1997; O'Hare, 2000; Gordon et al., 2005; Bridger, 2021; OpenText, 2022). In all these cases, the embeddings of documents from such divergent subdomains would be far away from one another in the embedding space when using standard RAG pipelines (as also confirmed by our own datasets in legal and plant accident use cases, see Section 5), underlying the relevance of MRAG.

# B  MATHEMATICAL FORMULATION: ADDITIONAL DETAILS

We omit, for clarity, unnecessary details such as layer normalizations. The output of attention head $h$ for the $i$th token $x_i$ is defined as follows (Vaswani et al., 2017):

$$\text{head}^h(\mathbf{x}_i) = \sum_j w_{ij} \mathbf{v}_j^h, \quad \text{where} \tag{1}$$

$$w_{ij} = \text{softmax}\left(\left(\mathbf{q}_i^h\right)^T \mathbf{k}_j^h\right), \quad \mathbf{q}_i^h = \mathbf{W}_q^h \mathbf{x}_i, \quad \mathbf{k}_j^h = \mathbf{W}_k^h \mathbf{x}_j, \quad \mathbf{v}_j^h = \mathbf{W}_v^h \mathbf{x}_j \tag{2}$$

where $\mathbf{W}_q^h, \mathbf{W}_k^h, \mathbf{W}_v^h$ are, respectively, learnable query, key, and value projections associated with head $h$, and $\mathbf{x}_j$ is the vector embedding of the $j$th token $x_j$. These outputs get combined to form the output of the $i$th multi-head attention block as follows:

$$\text{MHA}(\mathbf{x}_i) = \mathbf{W}_o \, \text{concat}(\text{head}^1(\mathbf{x}_i), \dots, \text{head}^h(\mathbf{x}_i))^T \tag{3}$$

where matrix $\mathbf{W}_o$ is the linear layer that combines the outcomes of all attention heads.

## B.1  STANDARD RAG EMBEDDING

In standard RAG, a single embedding vector $\mathbf{e}_{\text{std}} \in \mathbb{R}^d$ is computed for a chunk by extracting the decoder output for the final token $x_n$ after the final feed-forward layer:

$$\mathbf{e}_{\text{std}} = \text{FFN}\left(\text{MHA}(x_n)\right) \tag{4}$$

## B.2  MULTI-HEAD RAG EMBEDDING

Instead of compressing all head outputs into a single embedding, MRAG leverages the individual output of each head on the final token $x_n$:

$$\mathcal{S} = \{\mathbf{e}_k = \text{head}^k(\mathbf{x}_n) \in \mathbb{R}^{d/h} \mid k = 1, \dots, h\} \tag{5}$$

This results in $h$ head-wise embeddings per chunk, capturing diverse semantic aspects. Crucially, this design avoids any increase in memory or compute during inference, as head-level vectors are computed as part of the standard MHA process.

## C  System Design & Implementation: Additional Details

We provide addition details on system design and implementation.

### C.1  Ranking Strategy Details

The scoring of embedding spaces is detailed in Algorithm 1.

---

**Algorithm 1** Importance scores for heads.

---

**for** each head $h_i$ **do**
    $a_i, b_i, count\_a_i, count\_b_i \leftarrow 0$
    **for** each embedding $e_{ij}$ in $h_i$ **do**
        $a_i \leftarrow a_i + ||e_{ij}||$
        $count\_a_i \leftarrow count\_a_i + 1$
        **for** subset of $m$ embeddings $e_{ih}$ sampled uniformly at random **do**
            $b_i \leftarrow b_i + \text{cosine-distance}(e_{ij}, e_{ih})$
            $count\_b_i \leftarrow count\_b_i + 1$
        **end for**
    **end for**
    $a_i \leftarrow a_i/count\_a_i; b_i \leftarrow b_i/count\_b_i$
    $s_i \leftarrow a_i \cdot b_i$
**end for**

---

### C.2  Ranking Strategy Details

The voting strategy used by MRAG in its reranker is pictured in Algorithm 2.

---

**Algorithm 2** Voting strategy.

---

$l \leftarrow []$
**for** each head $h_i$ and its score $s_i$ **do**
    find best matching $c$ text chunks
    **for** each chunk $d_{i,p}$ with index $p$ in top $c$ **do**
        $w_{i,p} \leftarrow s_i \cdot 2^{-p}$
        add tuple $(d_{i,p}, w_{i,p})$ to $l$
    **end for**
**end for**
sort $l$ using weights $w_{i,p}$
return top $k$ elements of $l$

---

### C.3  Integration with Data Stores

MRAG can be seamlessly used with different classes of data stores ⬤ and nearest neighbor (NN) search approaches. It can be combined with both the exact and the approximate NN to find the matching (embedding, chunk)-pairs. These two parts of the broader RAG processing pipeline are orthogonal to MRAG.

# D  SPECIFICATION OF PROMPTS

## D.1  PROMPT TEMPLATE FOR READER

---
**Prompt Template for Reader**

You are presented with a series of articles, each potentially addressing different aspects of a topic.

1. <article title>
[<metadata>]
<body>
---------

2. <article title>
[<metadata>]
<body>
---------

<...>
---------

Task: Carefully analyze the articles. When formulating your response to the question below, identify the relevant aspects or claims made within each article. Construct your answer by comparing, contrasting, or synthesizing these points in a coherent and logically structured manner. Your response should be supported by specific references to the content of the articles. Where applicable, acknowledge differences in perspectives, data, or assumptions across the sources. Aim for clarity, precision, and concise reasoning grounded in the evidence provided.

Please answer the following question: <query text>

---

## D.2  PROMPT TEMPLATE FOR THE SYNTHETIC DATASET GENERATION

---
**Prompt Template for Query Generation**

Please create a story about the attached <number of articles> articles on the topics <list of titles>.

It is very important that each of the attached articles is relevant to the story, in a way that references the content of the article, not just its title. But please also mention each title at least once. Please make sure that all of the attached articles are relevant to your story, and that each article is referenced in at least two sentences! They do not necessarily have to be referenced in the same order, but make sure no article is forgotten.

Important: Output only the story, no additional text. And do not use bullet points, or paragraphs.

Articles:
---------
Article <title>:
<body>

<...>
---------
Again, make sure that you reference all the following topics in your story: <list of titles>

---

# E    MULTI-ASPECTUALITY WITH ATTENTION HEADS WITHOUT ADDITIONAL TRAINING

In MRAG, we extract embeddings from the hidden representations immediately after the attention block in the last decoder layer, avoiding any fine-tuning. This decision is based on an existing hypothesis that attention heads in Transformer models naturally differentiate during training, each attending to distinct aspects of the input data distribution.

## E.1    LITERATURE SURVEY

This hypothesis has been substantiated across various Transformer families. Wang et al. (2025a) introduce the *Local Learning Coefficient* (LLC), a measure of training dynamics at the head level. They show that attention heads begin with similar behavior but quickly diverge into functionally distinct clusters, each specializing in different patterns, ranging from local structure to multigram token groups. Olsson et al. (2022) identified "induction heads" in GPT-style models: specific heads that attend to earlier repeated sequences (e.g., in patterns like "X ... Y ... X"), enabling the model to learn simple in-context repetition *without additional supervision*. Further studies observed heads that consistently attend to names, suppress repetition, or shift attention predictably to structurally aligned tokens (McDougall et al., 2024; Gould et al., 2024).

Research in BERT-style models further supports these trends. Clark et al. (2019) demonstrate that different heads attend to direct objects, nominal modifiers, or punctuation tokens. Kovaleva et al. (2019) identify broad attention patterns like "vertical" heads (focusing on special tokens) and "diagonal" heads (attending to adjacent words). Htut et al. (2019) show that many heads correlate with syntactic dependency arcs.

Notably, while many heads specialize in meaningful ways, others appear to contribute little to model performance. Voita et al. (2019) and Michel et al. (2019) show that a large fraction of heads can be pruned without substantial performance loss, suggesting that specialization tends to concentrate in a smaller subset of effective heads.

## E.2    ANALYSIS OF MULTI-HEAD PATTERNS

We investigated the attention heads of two models in detail: LLaMA-2 7B and SFR-Embedding-Mistral. We selected these two models for a detailed investigation because the former represents models that are not fine-tuned for text embeddings, while the latter is specifically the text embedding model that we used for our experiments. For each model, we looked specifically at the attention scores within each attention head, i.e., how much attention each head pays to each input token during the inference. Knowing the semantics of the input tokens enables then deriving certain conclusions about multi-aspectuality and attention heads.

We plot selected results in Figure 7. Each heatmap shows the dot-product between key- and value-projections inside a given specified attention head, where line $i$ of a heatmap for attention head h indicates the dot-products between the query-projection of token $i$ and the key-projections of all previous tokens $j < i$ (both models use causal attention).

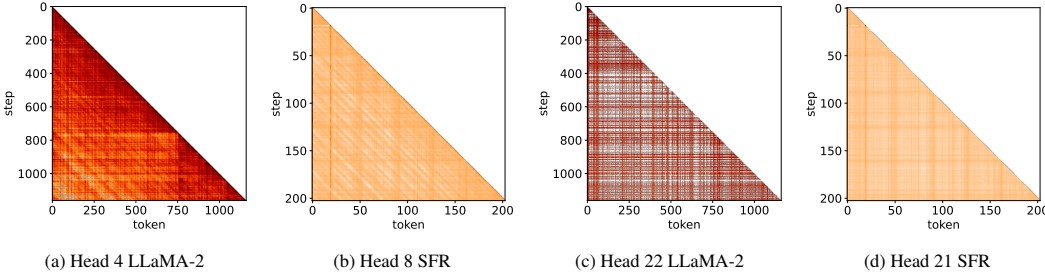

|        |       |        |         |
|--------|-------|--------|---------|
| (a) Head 4 LLaMA-2 | (b) Head 8 SFR | (c) Head 22 LLaMA-2 | (d) Head 21 SFR |

Figure 7: Heatmap plots for selected attention heads of the LLaMA-2 7B and SFR-Embedding-Mistral models.

For both models, we found out that the attention patterns vary significantly between the different attention heads. Still, we encountered two distinct patterns. First, the diagonal lines in Figures 7a and 7b indicate that, when processing a certain input token x, elevated attention is paid to some tokens that came a constant numbers of steps before x. We postulate that this pattern is likely beneficial to understanding the overall rhythm of a natural language, allowing the model to better

identify which words are semantically connected, and which parts of the input text refer to each other. Second, horizontal and vertical lines in Figures 7c and 7d show that these heads learned to pay attention to specific tokens, regardless of how far apart they are within the input sequence. An intuitive justification for such patterns is the focus on certain semantic aspects of the input sequence.

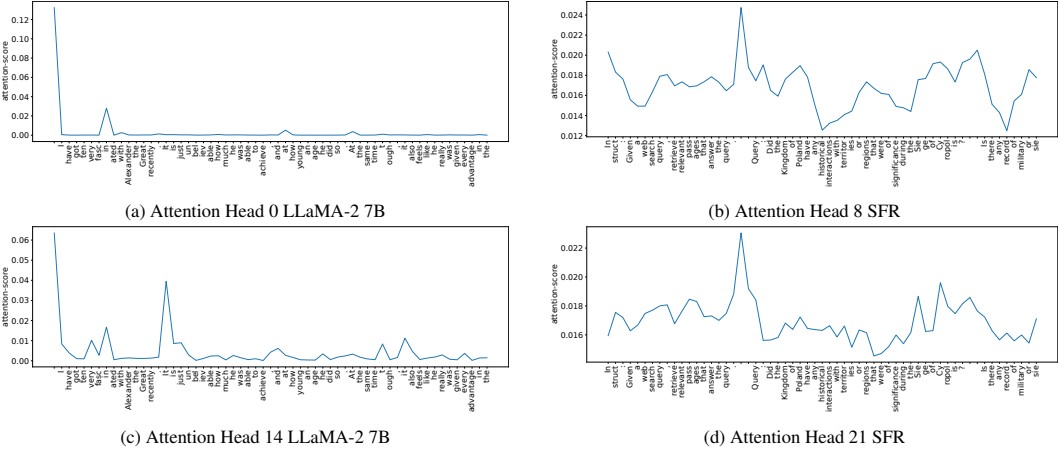

(a) Attention Head 0 LLaMA-2 7B  (b) Attention Head 8 SFR

(c) Attention Head 14 LLaMA-2 7B  (d) Attention Head 21 SFR

Figure 8: Attention scores for selected attention heads of the LLaMA-2 7B and SFR-Embedding-Mistral models.

We also detail attention scores (after applying softmax) of selected heads in Figures 8 and 9, when the model is processing the last token of its input. We see that some tokens gather a lot of attention from most heads, yet there is always a plethora of passages which are attended differently by any two attention heads. An interesting pattern we encountered was that for the SFR-Embedding-Mistral model (see Figure 9), all heads' attention spiked significantly on the first line-break in the input sequence - either positively or negatively. We conjecture that this is a consequence of how the embedding model was fine-tuned and its intended usage pattern: embedding queries are usually prepended with a retrieval instruction, which is terminated by a line-break. The model likely learnt to summarise the necessary information about this instruction inside the terminating line-break.

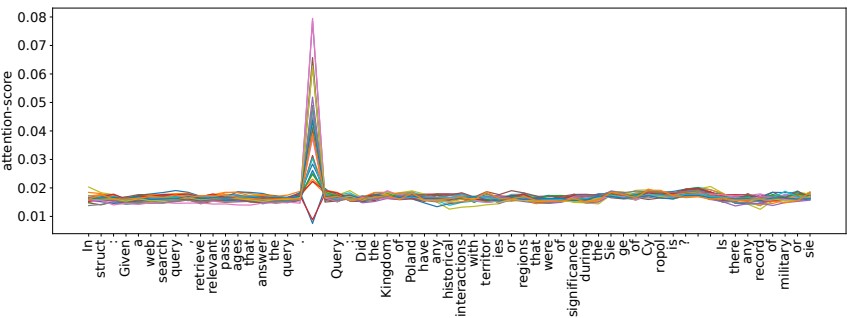

Figure 9: Attention scores for all attention heads of the SFR-Embedding-Mistral model.

# F  COMPLEXITY ANALYSIS: ADDITIONAL DETAILS

We now discuss additional details for our complexity analysis of the RAG schemes.

In Poly-encoders (Humeau et al., 2020) retrieval, we use $s(n)$ to represent additional attention evaluations needed to prepare the $m$ code embeddings and their final context embeddings with all $n$ documents.

In Lewis et al. (2020) retrieval, the work and depth are increased by the additional generator model evaluation.

In ColBERT (Khattab & Zaharia, 2020) retrieval, the work is increased linearly and depth logarithmically with the average query and document size due to the `maxsim` evaluations. For storage, the overhead scales linearly with the average size of the document, due to per-token embeddings.

In EMDR (Singh et al., 2021) retrieval, the work is increased $k+1$ times due to T5 encoders run on the top $k$ documents. This corresponds to a $D_m$ depth increase as the evaluations can be parallelized.

In Self-RAG (Asai et al., 2024), once the model decides to conduct retrieval using the `retrieval` token, it embeds the query, runs nearest $k$ search, and evaluates the model twice for each best-$k$ document to predict the `issup`, `isrel`, `isuse` labels. The depth increases only by $D_m$ as the evaluations can be parallelized.

In Chain-of-Note (Yu et al., 2024), during retrieval the $k$ best documents are fetched and then each is summarized sequentially by generating notes, resulting in a $ks(n)$ work and depth increase, with $s(n)$ representing the cost to generate an average note.

In RAPTOR (Sarthi et al., 2024), preprocessing involves creating a document tree, with leaves representing documents, and parents containing summaries of children, a cost we represent with $s(n)$ both for work and storage. Retrieval involves traversing the tree, which we encapsulate in a different $s(n)$.

RAGraph (Jiang et al., 2024) preprocessing involves embedding the documents by a graph model and creating the toy graphs, which we account for using $W_e$ and $D_e$. The toy graphs also create additional space requirements, which we account for using $s(n)$. For retrieval, additional key information such as environment or position-aware codes is used, represented by another $s(n)$ in our notation.

In RQ-RAG (Chan et al., 2024), the retrieval may be decomposed into multiple separate queries that are generated by the model and then evaluated in sequence using standard techniques, resulting in increases in work and depth that we denote using $s(n)$.

In ActiveRAG (Xu et al., 2024), retrieval is conducted using standard techniques, followed by three separate model evaluations: the Knowledge Assimilation (KA), Self-Inquiry (Q), and Thought Accommodation (TA) agents. As TA is similar to the work of an LLM answering the query, we omit its cost, like in all other frameworks, resulting in triple the work. Q and KA can be evaluated in parallel, increasing the depth only twice.

HiQA (Chen et al., 2024c) combines multiple retrieval strategies, using vector similarity matching, elastic search with BM25, and keyword matching. We represent these with $s(n)$ for work. Note that as these are independent, the depth is not increased. For preprocessing, HiQA uses a Hierarchical Contextual Augmentor, which creates a data hierarchy and introduces an overhead we denote by $s(n)$. HiQA also stores more information alongside vectors, such as keywords, increasing requirements by $s(n)$.

GraphRAG (Edge et al., 2025) creates a knowledge graph during preprocessing, which we denote with $W_e$ and $D_e$. It then extracts communities and their summaries, which we account for using $s(n)$. These are stored and require additional space also denoted by a different $s(n)$. During retrieval, GraphRAG uses an LLM to rank all communities in parallel by how useful they are which we estimate using $s(n)$.

In Fusion RAG (Rackauckas, 2024) retrieval, $k$ queries are generated and for each a standard RAG is evaluated, together with a reranking achieved by another model evaluation, which we estimate as $s(n)$.

In Meta-chunking (Zhao et al., 2025b), the documents are preprocessed by splitting them into chunks based on perplexity or margin sampling. As the decision whether to split or combine sentences in a document is based on an LLM, the preprocessing requires $l_d$ evaluations with some $s(n)$ postprocessing. The storage requirements are also increased by a different $s(n)$ as documents are stored not as a single vector but multiple vectors based on chunks. For the same reason, retrieval requires more work, as the number of vectors is increased by $s(n)$.

In MoC (Zhao et al., 2025a), chunks are created based on different granularities. Similarly to Meta-chunking, this means a larger number of vectors resulting in increased storage requirements, and retrieval work by $s(n)$. As MoC also includes routing and meta-chunkers, the preprocessing cost is increased by $s(n)$.

In Parametric RAG (Su et al., 2025), documents are represented as deltas of parameters that can be applied to the model. During preprocessing, the model is fine-tuned on a given document, resulting in a considerable increase of $s(n)$ in work and depth. As parameters are considerably larger than the hidden dimension, storage is also increased by a different $s(n)$. During retrieval, standard RAG fetches the documents, and the original model needs to be updated with their parameters, increasing work and depth by $s(n)$.

SuperRAG (Yang et al., 2025) first embeds the documents in a knowledge graph, which we represent by $W_e$ and $D_e$, and then uses this knowledge graph in retrieval to index it using $W_i$ and $D_i$. These also include any reranking SuperRAG might do.

HiRAG (Huang et al., 2025a) creates a hierarchical knowledge graph used for indexing in retrieval. Similarly, to SuperRAG, this increases the preprocessing and retrieval costs. We include in these the additional description and report generation that HiRAG conducts.

## G  FULL SPECIFICATION OF BENCHMARKING MULTI-ASPECTUALITY

We provide more details on how to benchmark multi-aspectuality in RAG. Figure 10 shows an example query and metrics usage. Each query requires retrieving a specific number of documents and the corresponding non-overlapping categories which define the ground truth. We fetch the top $k$ documents from a database, where $k$ is the "total number of documents fetched for a tested RAG scheme" (including potentially mismatches). Among these $k$ documents, we search for matches with the ground truth.

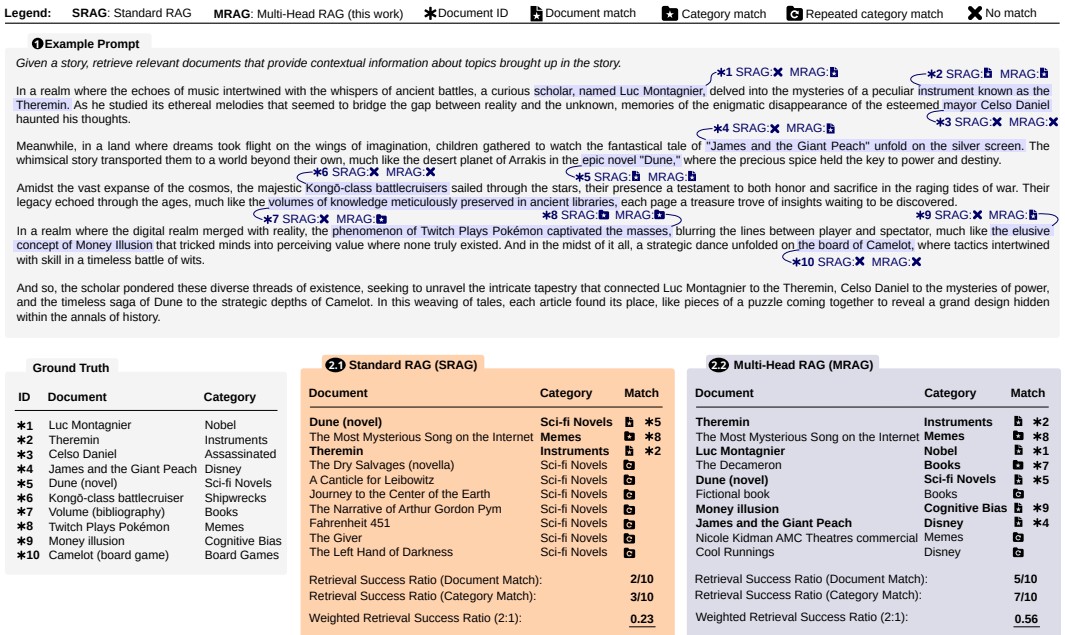

Figure 10: An example query used to evaluate different RAG strategies. We mention the documents to be fetched in the text and then assess the success ratio of different RAG strategies in finding these documents and their categories. We mark exact document matches 🗎, category matches 🗂, documents that match a category multiple times 🗂, and text segments with no matching document ✖. Finally, we show the weighted success ratio for each strategy, taking a 2:1 weighting (prioritizing the exact article matches).

### G.1  MULTI-ASPECT DATASETS

We first select conceptually different categories of documents for a synthetic dataset. Here, we harness publicly available Wikipedia articles. In the dataset construction pipeline, the user selects a given number of categories (e.g., countries, board games, historical swords, shipwrecks, etc.) and then, for each category, they sample a specified number of documents. The first part of the document (overview) is used as a text chunk to be embedded. We enforce that each overview must have at least 800 characters, matching commonly used chunk sizes in RAG schemes. We also use multi-aspect **real-world inspired datasets** consisting of NDAs and reports describing industry accidents in chemical processing plants. We ensure the usefulness of these datasets by working directly with tech leaders from 3 corporations that rely on RAG in their in-house LLM-driven report generation and analytics frameworks. Example categories of the legal documents are legal areas (energy law, family law, criminal law, etc.) or document language style (aggressive, mild, neutral, etc.). Examples of accident causes are natural disasters, human mistakes, or lack of proper training. We fully release these datasets to propel RAG research. Details on all three datasets can be found in the Appendix H.2. In our evaluation, we use a total of 16,500 documents.

### G.2  MULTI-ASPECT QUERY GENERATION

We also require queries that touch upon a given *number of $n$ aspects*. For example, a query with 10 aspects must contain a question about 10 different documents from 10 different categories. We create such queries by selecting $n$ categories, sampling a document from each selected category (ensuring there are no duplicates overall), and then generating a story that combines these documents, using an LLM (GPT-4o). We construct 160 queries with 1, 2, 3, 4, 5, 6, 10, 15, 20 and 25 aspects (1600 queries in total). An example multi-aspect query sent to the LLM that requires retrieving 10 documents from 10 different categories, is pictured in the top part of Figure 10.

### G.3 METRICS

We also design novel metrics to assess how well a given RAG scheme supports multi-aspectuality. For a query $Q$, a used Reranker scheme $S$ (detailed in Section 2.3), and $n$ documents from $n$ categories to retrieve, $Q_{rel}$ denotes the *ideal* set of documents that should be retrieved for $Q$. Then, $S(Q,n)$ is the set of the *actually* retrieved documents. We define the *Retrieval Success Ratio* as $\Xi(Q,n) = \frac{|S(Q,n) \cap Q_{rel}|}{|Q_{rel}|}$, i.e., the ratio of successfully retrieved relevant documents. Moreover, there is a case when a RAG scheme does not retrieve the *exact* desired document, but it still retrieves successfully *some other document* from *the same* category. While less desired, it still increases chances for a more accurate LLM answer following the retrieval. For example, when asking the LLM to determine the cause of an industry accident, fetching the documents in the same category as the accident being queried about, improves the chances for the LLM to give a more relevant answer. To consider such cases, we use another measure, the **Category Retrieval Success Ratio** or $\Xi_c$. It has the same form as $\Xi(Q,n)$ above, with one difference: $S(Q,n)$ is now the set of all the retrieved documents that belong to categories of the ideal desired documents. Finally, to combine these two metrics, we use the **Weighted Retrieval Success Ratio** $\Xi_w$ as $\Xi_w = \frac{w \cdot \Xi + \Xi_c}{w+1}$.

An example of using these metrics to assess how well MRAG and Standard RAG capture multi-aspectuality is pictured in the bottom part of Figure 10.

# H    EVALUATION SETUP: ADDITIONAL DETAILS

## H.1    COMPUTE RESOURCES

Our experiments were executed with compute nodes containing 4x NVIDIA GH200 and a total memory of 800 GB. In general one GPU with at least 40GB of memory should suffice. We used at most 50GB of storage and the OpenAI API as an external resource. The full experiments took at most three hours of GPU time and the cost for the OpenAI API were at most \$15. We carried out additional experiments, which amounted to around 20 hours of GPU time and cost of \$25 for the OpenAI API. Additional evaluation was executed with a mix of compute resources including NVIDIA A100 and V100 GPUs.

## H.2    DATASET DETAILS

Table 4: Overview of the structure and the number of documents in the respective datasets.

| dataset | #categories | #topics | #documents | total #documents |
|---------|-------------|---------|------------|------------------|
| Wikipedia | 80 | 50 documents per category | | 4000 |
| Legal Documents | 25 | 25 per category | 10 per topic | 6250 |
| Accident Reports | 25 | 25 per category | 10 per topic | 6250 |

# I EVALUATION: ADDITIONAL RESULTS

We provide additional empirical evaluation results.

## I.1 HARNESSING DIFFERENT DECODER BLOCKS

We analyze the impact of using embeddings from **different decoder blocks** for MRAG (instead of the last one). Here, we consider taking multi-aspect embeddings from three different layers of the embedding model: after the first multi-head attention block, after multi-head attention block 16 (in the middle of the decoder architecture), and the final multi-head attention. We discover that the last multi-head attention performs the best when compared with the Standard RAG.

## I.2 ANALYZING DIFFERENT VOTING STRATEGIES

We also illustrate selected representative data from a long investigation into two **additional voting strategies** for MRAG. We compare **MRAG (1)** where only the exponential lowering of significance of selected chunks is applied ($w_{i,p} = 2^{-p}$), and **MRAG (2)** which assigns the weight for each text chunk based on the distance between the particular text chunk ($d_{i,p}$) and the query ($q$) ($w_i = \frac{1}{distance(d_{i,p},q)}$). Figure 11 shows that these voting strategies perform worse on average than our selected strategy for MRAG, justifying its design and selection (described in Section 2.3).

We also consider two voting strategies for Split RAG, to further deepen the empirical evaluation. **Split (1)** only uses the exponential lowering of significance ($w_{i,p} = 2^{-p}$) and **Split (2)** which uses the same strategy as MRAG ($w_{i,p} = s_i \cdot 2^{-p}$). Figure 11 (on the right) shows that these voting strategies are on-par with each other while being worse than MRAG, further showcasing the advantages of MRAG.

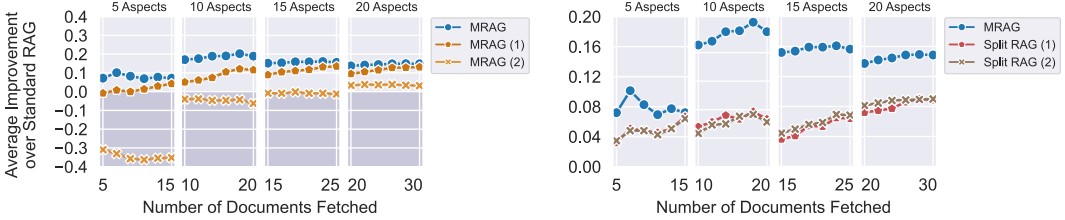

Figure 11: Evaluation of different voting strategies for MRAG and Split RAG.

## I.3 ANALYZING PREPROCESSING OVERHEAD

One-time head importance scoring in MRAG introduces minimal preprocessing overhead on top of the standard embedding scheme. The scoring consists of computing: (i) average L2 norms per embedding space, and (ii) average pairwise cosine distances among embeddings within each space. To assess practical overhead, we analyze six datasets using the SFR-Embedding-Model. The additional time required to compute importance scores is measured as a percentage of the original embedding time. For example, on SciFact, the overhead was just 2.7%, and on NFCorpus, only 1.75%. Even for moderate-scale corpora such as ArguAna (310 seconds total encoding time), the overhead remained under 5%. These results assume full pairwise distance computation across all chunks; in practice, the harnessed sampling makes them even lower.

